Genome-wide identification and expression analysis of GA20ox and GA3ox genes during pod development in peanut

Sun Jie 1 2
Zhang Xiaoqian 1 2
Fu Chun 3
Ahmad Naveed 4
Zhao Chuanzhi 1
Hou Lei 1
Naeem Muhammad 5
Pan Jiaowen 1
Wang Xingjun 1 2
Zhao Shuzhen zhaoshuzhen51@126.com 1 2
1 Institute of Crop Germplasm Resources, Shandong Academy of Agricultural Sciences; Shandong Provincial Key Laboratory of Crop Genetic Improvement, Ecology and Physiology , Jinan , China
2 College of Life Sciences, Shandong Normal University , Jinan , China
3 Weifang Academy of Agricultural Sciences , Weifang , China
4 Joint Center for Single Cell Biology, Shanghai Collaborative Innovation Center of Agri-Seeds, School of Agriculture and Biology, Shanghai Jiao Tong University , Shanghai , China
5 Department of Plant Science, School of Agriculture and Biology, Shanghai Jiao Tong University , Shanghai , China
Kutlu Imren
Electronic publication date: 2023 Oct 26
Publication date: 2023
Volume: 11
Electronic Location ID: e16279
Received 2023 Jun 5; Accepted 2023 Sep 20
Copyright: ©2023 Sun et al.
Copyright year: 2023
Copyright holder: Sun et al.
License: This is an open access article distributed under the terms of the Creative Commons Attribution License, which permits unrestricted use, distribution, reproduction and adaptation in any medium and for any purpose provided that it is properly attributed. For attribution, the original author(s), title, publication source (PeerJ) and either DOI or URL of the article must be cited.
License URL: https://creativecommons.org/licenses/by/4.0/

Keywords: Arachis hypogaea, Gibberellin biosynthesis, Gene expression analysis, Pod development

Funding: Key Research and Development Project of Shandong Province 2022LZGC007 Agricultural scientific and technological innovation project of Shandong Academy of Agricultural Sciences CXGC2023C04 National Natural Science Foundation of China 31861143009 32072090 Taishan Scholar Project of Shandong Province ts20190964 This work was supported by the Key Research and Development Project of Shandong Province (2022LZGC007), the Agricultural scientific and technological innovation project of Shandong Academy of Agricultural Sciences (CXGC2023C04), the National Natural Science Foundation of China (31861143009; 32072090) and the Taishan Scholar Project of Shandong Province (ts20190964). The funders had no role in study design, data collection and analysis, decision to publish, or preparation of the manuscript.

==============================
Background

Gibberellins (GAs) play important roles in regulating peanut growth and development. GA20ox and GA3ox are key enzymes involved in GA biosynthesis. These enzymes encoded by a multigene family belong to the 2OG-Fe (II) oxygenase superfamily. To date, no genome-wide comparative analysis of peanut AhGA20ox and AhGA3ox-encoding genes has been performed, and the roles of these genes in peanut pod development are not clear.

Methods

A whole-genome analysis of AhGA20ox and AhGA3ox gene families in peanut was carried out using bioinformatic tools. The expression of these genes at different stage of pod development was analyzed using qRT-PCR.

Results

In this study, a total of 15 AhGA20ox and five AhGA3ox genes were identified in peanut genome, which were distributed on 14 chromosomes. Phylogenetic analysis divided the GA20oxs and GA3oxs into three groups, but AhGA20oxs and AhGA3oxs in two groups. The conserved pattern of gene structure, cis-elements, and protein motifs further confirmed their evolutionary relationship in peanut. AhGA20ox and AhGA3ox genes were differential expressed at different stages of pod development. The strong expression of AhGA20ox1/AhGA20ox4, AhGA20ox12/AhGA20ox15, AhGA3ox1 and AhGA3ox4/AhGA3ox5 in S1-stage indicated that these genes could have a key role in controlling peg elongation. Furthermore, AhGA20ox and AhGA3ox also showed diverse expression patterns in different peanut tissues including leaves, main stems, flowers and inflorescences. Noticeably, AhGA20ox9/AhGA20ox11 and AhGA3o4/AhGA3ox5 were highly expressed in the main stem, whereas the AhGA3ox1 and AhGA20ox10 were strongly expressed in the inflorescence. The expression levels of AhGA20ox2/AhGA20ox3, AhGA20ox5/AhGA20ox6, AhGA20ox7/AhGA20ox8, AhGA20ox13/AhGA20ox14 and AhGA3ox2/AhGA3ox3 were high in the flowers, suggesting their involvement in flower development. These results provide a basis for deciphering the roles of AhGA20ox and AhGA3ox in peanut growth and development, especially in pod development.

Introduction

Peanut (Arachis hypogaea L.), rich in protein, oil, vitamins and other nutrients, is one of the top five oil crops in the world (Toomer, 2018). Pod directly determines the yield and quality of peanut. Interestingly, a unique feature of pod development in peanut is that the plants flower and fertilize above-ground while pod forms and matures under the ground. After flowering and fertilization, the fertilized egg only divides several times to form the pro-embryo and then stops dividing. While the stalk of the ovary extends continuously with the unexpanded ovary after fertilization to form a peg growing toward the ground. The peg expands horizontally after the ovules are buried in the soil (Zhang et al., 2016). The development of peanut pod is regulated by a variety of hormones such as auxin, gibberellins (GAs), brassinosteroids (BR), abscisic acid (ABA), ethylene and cytokinin. In the early stage of peanut pod development, auxin and GAs promote the elongation and growth of peg, while cytokinin regulates the cell division of peg (Edgar, 2003). GAs are widely involved in various stages of plant growth and development (Binenbaum, Weinstain & Shani, 2018), such as seed germination (Xu et al., 2020), stem elongation (Zhang et al., 2021), shade response (Yang & Li, 2017), flowering (Bao et al., 2020), and fruit development (Hu et al., 2018). So far, 136 GAs with definite structures have been identified in plants, bacteria and fungi. However, only a few GAs have physiological effects on plant growth and development, such as GA1, GA3, GA4 and GA7 (Giacomelli et al., 2013; MacMillan, 2001).

The GA20 oxidase (GA20ox) and GA3 oxidase (GA3ox) are key rate-limiting enzymes in GA biosynthesis to produce bioactive GAs. At the later stage of GA biosynthesis, GA20ox catalyze inactive GA12 and GA53 into GA9 and GA20 by removing the C-20. Subsequently, GA9 and GA20 were catalyzed by GA3ox to produce bioactive GA4 and GA1 through 3β-hydroxylation (Hedden, 2020; Sonia et al., 2018). GA20ox and GA3ox are the members of 2OG-Fe (II) oxygenase superfamily and generally encoded by multiple genes in plants. Rice sd1 (semi-dwarf 1) gene encodes GA20ox2 and mutation of this gene can inhibit GA biosynthesis and lead to semi-dwarf phenotype (Sasaki et al., 2002). The ga3ox1 mutation in Arabidopsis thaliana led to semi-dwarf, and GA3ox1 and GA3ox2 double mutants showed more sever phenotype than the single mutant (Mitchum et al., 2006).

Fruit setting is a key process in agricultural production and is usually triggered by ovule fertilization (Ozga et al., 2009). Plant hormones play an important role in fruit set and development. IAA and GA are the main hormones promoting fruit development (Liu et al., 2018). DELLA protein is a key negative regulator of GA signaling pathway and acts as a fruit growth inhibitor prior to fertilization. In pollinated ovaries, the increased GA content leads to DELLA degradation through 26S proteasome pathway, thus releasing the inhibitory effect of DELLA (Fuentes et al., 2012). In addition, exogenous application of GAs, independent of fertilization, could promote fruit-setting and development in pears, apricots, strawberries and grapes. After GA3 treatment, the expressions of ARF2 and ARF8 were inhibited, suggesting that GA-induced parthenogenesis might be caused by the downregulation of ARF2 and ARF8 (Maaike, Mariani & Vriezen, 2009). However, the genome-wide identification of the AhGA20ox and AhGA3ox families in peanut have not been reported. The functions of GA20ox and GA3ox family genes in peanut pod development are not clear.

In the current study, all members of the AhGA20ox and AhGA3ox gene families were identified in peanut genome. The phylogenetic analysis of these genes was carried out. The expression patterns of AhGA20ox and AhGA3ox genes at different stages of peanut pod development were examined by qRT-PCR. These results provided valuable information for us to understand the role of these genes in peanut pod development.

Materials and Methods

Identification of AhGA20ox and AhGA3ox gene family members in peanut

All sequences were downloaded from four databases: TAIR (Arabidopsis Information Resource, http://www.arabidopsis.org/), Rice (Rice Information Resource, http://www.rice.plantbiology.msu.edu/), Soybean Genome Annotation Project Database (http://www.phytozome.net/soybean/) and NCBI Genome Database (https://www.ncbi.nlm.nih.gov/). The Arabidopsis GA20ox and GA3ox protein sequences were first downloaded from TAIR website. Subsequently, the amino acid sequences of GA20oxs and GA3oxs of Arabidopsis were searched against peanut genome using NCBI database. The resulting sequences were further validated in SMART (http://smart.embl.de) and Pfam (http://pfam.xfam.org) to obtain all members of AhGA20ox and AhGA3ox with the 2OG-Fe (II) oxygenase domain (PF03171) in peanut genome. Previous study showed that the allotetraploid A. hypogaea (AABB, 2n = 4x = 40) is from hybridization between diploids A. duranensis (AA genome) and A. ipaensis (BB). One to ten (A01–A10) chromosomes originate from A subgenome, and eleven to twenty (B01–B10) chromosomes originate from B subgenome. In this study, gene members were ordered according to the order of chromosomes numbering from A and B subgenome. For example, AhGA20ox1 and AhGA20ox2 are from chr. 2 (A02), while AhGA20ox3 and AhGA20ox4 are from corresponding chr.12 (B02) (Table 1). The isoelectric point (pI), molecular weight (MW), and amino acid (aa) number of GA20ox and GA3ox proteins were predicted using the ProtPram tool in ExPASy (https://www.expasy.org/).

Table 1 Physicochemical properties of AhGA20ox and AhGA3ox family members in peanut.

Gene name	Gene accession number	Chrome	Length of CDS	Length of peptide	PI	MW (Da)	
AhGA20ox1	NC_037619.1	2	1119	372	6.14	42612.51	
AhGA20ox2	NC_037619.1	2	1119	372	5.70	42603.44	
AhGA20ox3	NC_037629.1	12	1119	372	5.70	42531.33	
AhGA20ox4	NC_037629.1	12	1287	428	6.20	49169.01	
AhGA20ox5	NC_037620.1	3	1140	379	6.16	42955.71	
AhGA20ox6	NC_037630.1	13	1140	379	5.73	42927.69	
AhGA20ox7	NC_037621.1	4	1140	379	6.76	43758.31	
AhGA20ox8	NC_037631.1	14	1140	379	7.09	43671.10	
AhGA20ox9	NC_037622.1	5	1149	382	5.80	43477.38	
AhGA20ox10	NC_037622.1	5	1116	371	5.14	42256.14	
AhGA20ox11	NC_037632.1	15	1149	382	6.27	43374.34	
AhGA20ox12	NC_037625.1	8	1092	363	6.94	41469.38	
AhGA20ox13	NC_037626.1	9	1137	378	6.16	42856.01	
AhGA20ox14	NC_037636.1	19	1137	378	6.16	42909.95	
AhGA20ox15	NC_037634.1	17	1092	363	6.94	41371.32	
AtGA20ox1	NC_003075.7	4	1428	377	5.77	43224.26	
AtGA20ox2	NC_003076.8	5	1358	378	4.90	33520.78	
AtGA20ox3	NC_003076.8	5	1323	380	6.90	43437.39	
AtGA20ox4	NC_003070.9	1	1568	376	7.14	43133.35	
AtGA20ox5	NC_003070.9	1	1810	385	8.04	43161.15	
GmGA20ox1	NC_016090.4	3	1149	383	6.15	43484.61	
GmGA20ox2	NC_038243.2	7	1191	397	6.50	44987.53	
GmGA20ox3	NC_038245.2	9	1149	383	5.63	43363.39	
GmGA20ox4	NC_038246.2	10	1098	366	5.64	41163.03	
GmGA20ox5	NC_038249.2	13	1128	376	6.27	42797.56	
GmGA20ox6	NC_038250.2	14	1047	349	6.37	39525.06	
GmGA20ox7	NC_038252.2	16	1026	342	5.85	38535.98	
GmGA20ox8	NC_038256.2	20	1152	384	5.85	43327.47	
OsGA20ox1	NC_029258.1	3	1854	372	5.98	42255.71	
OsGA20ox2	NC_029256.1	1	3123	389	5.73	42513.19	
OsGA20ox3	NC_029262.1	7	2744	367	5.75	40494.75	
OsGA20ox4	NC_029260.1	5	6929	444	6.70	47634.91	
OsGA20ox5	NC_029258.1	3	2008	352	5.17	39293.06	
OsGA20ox6	NC_029259.1	4	2123	300	5.51	32102.33	
OsGA20ox7	NC_029263.1	8	2380	383	5.96	41831.59	
OsGA20ox8	NC_029259.1	4	3640	326	5.34	35797.44	
AhGA3ox1	NC_037624.1	7	1128	375	8.11	41693.86	
AhGA3ox2	NC_037625.1	8	1059	352	7.30	40020.57	
AhGA3ox3	NC_037635.1	18	1065	354	6.49	40322.80	
AhGA3ox4	NC_037626.1	9	1086	361	8.07	40659.64	
AhGA3ox5	NC_037636.1	19	1095	364	8.08	41025.03	
AtGA3ox1	NC_003070.9	1	1077	358	6.34	40161.81	
AtGA3ox2	NC_003070.9	1	1044	347	6.56	38782.41	
AtGA3ox3	NC_003075.7	4	1050	349	6.16	39210.76	
AtGA3ox4	NC_003070.9	1	1068	355	5.49	39152.51	
GmGA3ox1	NC_016091.4	4	1026	342	6.62	38420.91	
GmGA3ox2	NC_038242.2	6	1044	348	6.18	39059.39	
GmGA3ox3	NC_038249.2	13	1059	353	8.52	39245.13	
GmGA3ox4	NC_038250.2	14	1044	348	6.45	38768.46	
GmGA3ox5	NC_038251.2	15	1062	354	7.72	39313.07	
GmGA3ox6	NC_038253.2	17	1053	351	6.65	39126.85	
OsGA3ox1	NC_029260.1	5	1155	385	5.95	41555.12	
OsGA3ox2	NC_029256.1	1	1122	374	6.47	40572.22	

Phylogenetic analysis of GA20ox and GA3ox

Multiple sequence alignment of AhGA20ox and AhGA3ox amino acid sequences with the 2OG-Fe (II) oxygenase domain from peanut and homologous sequences from Arabidopsis thaliana, rice, and soybean was performed using the DNAMAN software (Vers7; Lynnon Corporation, Montreal, QC, Canada). Then, phylogenetic analysis was performed using AhGA20ox and AhGA3ox protein sequences together with other plant species by using neighbor-joining method in MEGA (Vers7; Pennsylvania State University, PA, USA) with bootstrap value of 1000 (Kumar, Stecher & Tamura, 2016). Finally, iTOL (https://itol.embl.de/) online software was used for the display, annotation and management of phylogenetic trees.

Chromosomal distribution and gene structure analysis of AhGA20ox and AhGA3ox

The loci of GA20ox and GA3ox genes were downloaded from the genome annotation file from Peanutbase in order to obtain chromosome location information. Then, the chromosome map was generated using Mapchart software (version 2.2) (http://mg2c.iask.in/mg2c_v2.1/). The online tool GSDS (version 2.0) (http://gsds.cbi.pku.edu.cn) was employed to analyze the genomic sequences of AhGA20ox and AhGA3ox for gene structure analysis.

Analysis of the conserved protein motifs and cis-acting elements

The online webserver of MEME software was used to analyze the conserved protein motifs of peanut AhGA20ox and AhGA3ox family members. The occurrence of top 8 conserved protein motifs in AhGA20ox and AhGA3ox sequences were further screened and analyzed.

The 2000 bp sequence upstream of the start codon of the AhGA20ox and AhGA3ox genes was downloaded from the peanut genome database (https://www.ncbi.nlm.nih.gov/), and the cis-acting elements were predicted and analyzed by PlantCARE online (http://bioinformatics.psb.ugent.be/).

AhGA20ox and AhGA3ox protein interaction networks prediction

The online webtool of STRING network (https://string-db.org/) was utilized to predict the functional protein interaction network of AhGA20ox and AhGA3ox. All possible interacting proteins constituting a hierarchical network with AhGA20ox and AhGA3ox were further classified and shown graphically in STRING-generated network.

Plant materials and treatment conditions

The peanut cultivar Jihua8 with high oleic acid developed by our lab was used in this study. Jihua8 plants were grown in the field condition from May to September 2022 at Jiyang Experimental Station of Shandong Academy of Agricultural Sciences, Shandong, China (36°58′34.53″N, 116°59′1.29″E). In the growing period, the temperature ranged from 20 °C to 37 °C. Stems, leaves, flowers, inflorescences and pods at different stages of development were collected with six biological replicates from plants. Pegs and pod samples were collected at 0, 3, 9, 12, 20, and 30 days after soil penetration (DAP), representing the S1, S2, S3, S4, S5, and S6 stages, respectively (Fig. 1). All the materials were frozen immediately in liquid nitrogen and stored at −80 °C for RNA extraction and gene expression analysis.

Figure 1 The phynotype of peg and pod at different developmental stages.

RNA extraction and expression analysis of AhGA20ox and AhGA3ox

Total RNA was extracted by using FastPure Plant Total RNA isolation kit (AG RNAex Pro Reagent, Changsha, China). The extracted RNA was used as template for reverse transcription reaction using Hiscrip II Q RT SuperMix for qRT-PCR (Vazyme, Nanjing, China). The primers of qRT-PCR were designed by Primer Premier software (Premier Biosoft International, Palo Alto, CA, USA) (Table S1). An ABI 7500 real-time PCR instrument (Thermo Fisher Scientific, Waltham, USA) was used for subsequent quantitative reaction. The reaction mixture consisted of 10 µL ChamQ SYBR qPCR Master Mix (Vazyme, Nanjing, China), 0.5 µL of each primer (10 µM), 2 µL cDNA template, and 7 µL RNase-free H2O. Reaction conditions included pre-denaturation (95 °C, 30 s), cyclic reaction (95 °C, 10 s; 60 °C, 30 s; 40 cycles). The relative expression level of these genes was analyzed by the 2−ΔΔCT method (Livak & Schmittgen, 2001), using Ahactin gene as the internal control.

Results

Identification and characterization of AhGA20ox and AhGA3ox genes

BLASTP (e-value ≦ 0.001) search was performed using the amino acid sequences of five Arabidopsis GA20ox and four GA3ox. After removing redundant sequences and confirming the presence of gibberellin-dioxygenases domains by SMART and Pfam, 20 gibberellin-dioxygenases genes including 15 GA20ox (AhGA20ox1-15) and five GA3ox (AhGA3ox1-5) were finally retained and used for further analysis. Further analysis showed that the length of amino acid encoded by AhGA20ox genes varied from 363 aa to 428 aa, and the molecular weight was 41.3 kDa-49.2 kDa. The isoelectric point (pI) values were between 5.14 and 7.09. The amino acid length of GA3oxs varied from 352 aa to 375 aa, the molecular weight was 40 kDa–41.7 kDa, and the pI values varied from 6.49 to 8.11 (Table 1).

Phylogenetic analysis of AhGA20ox and AhGA3ox

In order to analyze the phylogenetic relationship of GA20ox and GA3ox gene family, the phylogenetic tree was constructed using the protein sequences of GA20ox and GA3ox from peanut, Arabidopsis thaliana, rice, and soybean (Table S2). The results showed that the GA20oxs and GA3oxs were divided into three groups (group A-C). The largest group was group C containing 32 members of GA20ox and all peanut GA20oxs were in this group. Groups A was the smallest one only containing four members of rice GA20ox (Fig. 2). Group B contained 17 members, all of which were GA3ox. Moreover, in each group, genes from the same species showed higher similarity than those from different species. Further analysis revealed that GA20ox and GA3ox family members in peanut were closely related to those in soybean, but distantly related to those in rice (Fig. 2).

Figure 2 Phylogenetic analysis of GA20ox and GA3ox protein in Arabidopsis Thaliana (At), rice (Os), soybean (Gm) and peanut (Ah).

The phylogenetic tree was constructed using neighbor-joining method in MRGA 7.0 with bootstrap value of 1000. The numbers represent the scale of the evolutionary tree. Gene accession numbers of the sequences used in this tree are listed in Table S2.

Gene structure and conserved motifs of AhGA20ox and AhGA3ox genes

The exon-intron structure analysis of 15 AhGA20ox genes showed that AhGA20ox genes contained three exons and two introns except that AhGA20ox4 had four exons and three introns. The closer genes in the phylogenetic tree showed more similarity in structure, for example, AhGA20ox13/14 and AhGA20ox9/11. However, AhGA20ox4 and AhGA20ox1 were closely related in the phylogenetic tree, but very different in gene structures. It may due that AhGA20ox4 gene acquired a long intron in the first exon. Five AhGA3ox genes contained two exons and two introns, and the length of the introns varied greatly. These differences in exon-intron structure may suggest the functional differentiation of these genes (Fig. 3).

Figure 3 Gene structure organization of peanut GA20ox and GA3ox family members.

Exons (CDS) and UTR are represented by yellow boxes and blue boxes, respectively, and grey lines between exons represents introns.

The conserved motifs of AhGA20ox and AhGA3ox proteins were predicted. In the predicted eight motifs, the E value of each motif was significant, and the length of motif was 36-50 aa. Motif 1, 2, 3 and 5 were distributed in all AhGA20ox and AhGA3ox sequences (Fig. 4), suggesting that these four motifs may be the core conserved domain of these gene families. This indicated that the AhGA20ox and AhGA3ox families were highly conserved and presumably had some degree of functional redundancy. In addition, motif 8 existed only in the AhGA3ox gene family (Fig. 4), while motif 4 and 6 existed only in the AhGA20ox gene family (Fig. 4).

Figure 4 Distribution of conserved motifs in GA20ox and GA3ox family members.

The sequence information of motifs marked different colors is represented at the bottom.

Cis-element analysis of AhGA20ox and AhGA3ox genes

The analysis of cis-elements was performed using the 2000 bp promoter region of AhGA20ox and AhGA3ox genes. The results showed the presence of widely known eight key cis-elements in the promoter of these genes (Fig. 5, Tables S3 and S4). The most abundantly presented cis-elements mainly included light-responsive units such as Box 4, G-box, TCT-motif, GA-motif, GATA-motif, AT1-motif and GT1-motif. Hormone-responsive units such as ABA responsive-motif (ABRE), auxin-responsive element (TGA-element), gibberellin-responsive element (TATC-box, GARE-motif and P-box) and salicylic acid responsiveness (TCA-element) were also detected. Low-temperature responsive units (LTR), drought-inducibility (MBS) and defense and stress responsiveness (TC-rich repeats) elements were also identified. The occurrence of these cis-elements suggested that AhGA20ox and AhGA3ox genes are strongly linked to plant growth, development, and tolerance to varied stresses, as well as other crucial signaling pathways in peanut.

Figure 5 The organization of cis-acting elements in the promoter region of AhGA20ox and AhGA3ox genes in peanut.

Different colors were used to indicate different elements.

Chromosome location and selective pressure analysis of AhGA20ox and AhGA3ox genes

Based on the peanut genome information, AhGA20ox and AhGA3ox genes were distributed on 14 chromosomes. AhGA20ox1 and AhGA20ox2 were localized at chr.2. AhGA20ox9 and AhGA20ox10 were localized at chr.5, whereas AhGA20ox12 and AhGA3ox2 were mapped to chr.8. AhGA20ox13 and AhGA3ox4 were positioned at chr.9, while AhGA20ox3 and AhGA20ox4 were found at chr.12. AhGA20ox14 and AhGA3ox5 were located at chr.19. AhGA20ox5, AhGA20ox7, AhGA3ox1, AhGA20ox6, AhGA20ox8, AhGA20ox11, AhGA20ox15 and AhGA3ox3 were localized at chr.3, chr.4, chr.7, chr.13, chr.14, chr.15, chr.17 and chr.18, respectively. Notably, most of AhGA20ox and AhGA3ox genes located at the distal ends of chromosomes (Fig. 6, Table S5).

Figure 6 Chromosome mapping of AhGA20ox and AhGA3ox genes.

Fifteen AhGA20ox and five AhGA3ox genes were unevenly distributed on the 14 chromosomes, with the exception of chr. 01, 06, 10, 11, 16 and 20. The location on the chromosome of each AhGA20ox and AhGA3ox gene was indicated on the right side of the respective chromosome. The scale bar for chromosome length was showed at the left of all chromosomes.

TBtools was used to calculate the Nonsynonymous (Ka)/synonymous (Ks) ratios for each gene pair, to explore the evolutionary constraints of the peanut GA20ox and GA3ox genes (Table S6). The Ka/Ks ratios of gene pairs from peanut (Ah-Ah), soybean (Gm-Gm), and peanut and soybean (Ah-Gm) were all less than 1, indicating that the orthologs in the related species experienced purification selection, meanwhile, the paralogs in the species were highly conserved.

Interactive protein network of AhGA20ox and AhGA3ox proteins

The protein–protein interaction (PPI) network of AhGA20ox and AhGA3ox was investigated using the STRING database. The major interacting partners of AhGA20ox AhGA3ox were predicted as Fe2OG dioxygenase domain containing protein, which is a key component of iron-ascorbate dependent oxidoreductase family. Fe2OG enzyme is known to catalyze a wide range of oxidative reactions crucial to plant metabolisms. The interactions between GA20ox, GA3ox and GA2ox in peanut were very close. The lines in the figure indicated the confidence of the interaction between the two proteins, and the line thickness indicates the strength of data support. GO classification results indicated that these genes were mainly involved in short-day photoperiodism, flowering, GA metabolism, and GA-mediated signal transduction pathway. A0A444XWP4 had 18 interacting proteins, GA3ox1/4/5, A0A444Y5G9, A0A445BYM0 and A0A444YUS4 all had 17 interacting proteins (Table S7). A0A444XWP4, A0A444Y5G9 and A0A445BYM0 were speculated as GA2ox, a key enzyme catalyzing GA degradation. Amino acid sequences blasting result indicated that A0A444YUS4 was GA3ox. No interacting protein was found in AhGAox5/6/13/15, while AhGAox12 only had a weak interaction with A0A444XWP4. (Fig. 7). The prediction of PPI network of these GA biosynthesis enzymes in peanut provided important information for understanding the regulatory mechanism underlying gibberellin biosynthesis.

Figure 7 The prediction of protein–hprotein interaction network of AhGA20ox and AhGA3ox encoding proteins.

Nodes represent proteins, and lines indicate that they have interaction relationship between proteins.

Expression analysis of AhGA20ox and AhGA3ox genes during pod development

The expressions of GA20ox and GA3ox genes during peanut pod development were investigated using qRT-PCR. Peanut pod development was divided into six stages in this study. In S1 stage, the green or purple pegs were aerial-grown. In S2 stage, pegs had been embedded in the soil for 3 days and the color of the pegs is white. Pod enlargement in S2 pegs was not detected. In S3 stage, pegs had been buried in the soil for 9 days and pod enlargement was initiated. In S4 stage, pegs had been buried in the soil for 12 days. In S5 stage, pegs had been buried in the soil for 20 days. In S6 stage, pegs had been buried in the soil for 30 days (Fig. 1).

Both AhGA20ox and AhGA3ox gene families were expressed ubiquitously at different stages of pod development, but showed different expression patterns. The expressions of AhGA20ox2/AhGA20ox3, AhGA20ox6/AhGA20ox5, AhGA20ox7/AhGA20ox8 GA20ox8 and GA20ox10 gradually increased with the development of pod, and reached the highest level at S6 (Fig. 8, Table S8). The expression levels of AhGA20ox15/AhGA20ox12 and AhGA3ox1 gradually decreased with the development of peanut pod.Theses results indicated that different members of AhGA20ox and AhGA3ox families had different functions in pod development. AhGA20ox1/AhGA20ox4, AhGA20ox12/AhGA20ox15, AhGA3ox1 and AhGA3ox4/AhGA3ox5 showed high expression levels in S1 (Fig. 8), suggesting their possible roles in regulating peg elongation. Other AhGA3ox genes may be involved in regulating the expansion and growth of pod, even though they showed different expression patterns during pod development. For example, AhGA3ox1 and AhGA3ox2/AhGA3ox3 showed high expression levels in S2, and AhGA20ox13/AhGA20ox14 showed the highest expression level in S3. The expression levels of AhGA20ox1/AhGA20ox4 and AhGA20ox9/11 were the highest in S4 (Fig. 8, Table S8).

Figure 8 Expression analysis of AhGA20ox and AhGA3ox genes in different stages of peanut pod development.

The heatmap was generated with the qRT-PCR values of 15 AhGA20ox and 5 AhGA3ox genes using the online tool, TBtools, and the color scale beside the heat map indicates gene expression levels, low transcript abundance indicated by blue color and high transcript abundance indicated by red color. Fifteen AhGA20ox and 5 AhGA20ox genes were classifed into three groups Group I, AhGA20ox15/12/4/1 and AhGA3ox1/4/5; Group II, AhGA20ox2/3/5/6/10; Group III, AhGA20ox7/8/9/11/13/14 and AhGA3ox2/3.

Expression analysis of AhGA20ox and AhGA3ox genes in different tissues

The expressions of AhGA20ox and AhGA3ox genes in peanut leaves, main stems, flowers and inflorescences were analyzed. The results showed that AhGA20ox and AhGA3ox were expressed in different tissues with varied levels. The expression levels of AhGA20ox9/AhGA20ox11 and AhGA3ox4/AhGA3ox5 were the highest in the main stem, while the expression levels of AhGA20ox12/AhGA20ox15 were the lowest in the main stem (Fig. 9, Table S9). Except AhGA20ox9/AhGA20ox11, AhGA3ox1 and AhGA3ox4/AhGA20ox5, other genes showed high expression levels in flowers (Fig. 9), implying that these genes may be involved in the regulation of flower development. AhGA20ox2/AhGA20ox3, AhGA20ox5/AhGA20ox6, AhGA20ox7/AhGA20ox8, AhGA20ox13/AhGA20ox14 and AhGA3ox2/AhGA3ox3 were highly expressed in inflorescence (Fig. 9). AhGA20ox1/ AhGA20ox4, AhGA20ox12/AhGA20ox15 and AhGA3ox1 showed high expression levels in leaves (Fig. 9, Table S9).

Figure 9 Expression analysis of AhGA20ox and AhGA3ox genes in different tissues of peanut plants.

The heatmap was generated with the qRT-PCR values of 15 AhGA20ox and five AhGA3ox genes using the online tool, TBtools, and the color scale beside the heat map indicates gene expression levels, low transcript abundance indicated by blue color and high transcript abundance indicated by red color. Fifteen AhGA20ox and five AhGA3ox genes were classifed into two groups Group I, AhGA20ox9/11 and AhGA3ox1/4/5; Group II, AhGA20ox1-AhGA20ox9, AhGA20ox10, AhGA20ox12-AhGA20ox15 and AhGA3ox2/3.

Discussion

Gibberellins regulate plant growth and development and the enzymes encoded by GA20ox and GA3ox genes play key role in GA biosynthesis. GA20ox and GA3ox genes have been identified in many higher plants, for example, nine members in Arabidopsis (Han & Zhu, 2011), 10 members in rice (Han & Zhu, 2011), 14 members in soybean (Han & Zhu, 2011), 13 members in grape (He et al., 2019), nine members in Phyllostachys edulis (Ye et al., 2019) and 14 members in maize (Ci et al., 2021). In this study, a total of 15 AhGA20ox and five AhGA3ox family members were identified from peanuts. The evolutionary relationship of these genes in Arabidopsis, rice, soybean and peanut showed that functionally different GA3ox and GA20ox clustered in separate groups. AhGA20ox and AhGA3ox genes in peanut were more closely related to those in soybean, while far related to those in rice. In addition, the DIOX_N and 2OG-FeII_Oxy superfamily domains contained in the protein sequences of GA20ox and GA3ox are conserved domains shared by all species (Honi et al., 2020).

SD1 (OsGA20ox2) is a gene of the rice Green Revolution. It was found that the mutant of this gene causes semi-dwarf phenotype. It catalyzes the conversion of GA53, a precursor of gibberellin synthesis, to GA20 (Sasaki et al., 2002). In this study, AhGA20ox14, AhGA20ox9 and AhGA20ox11 showed the range of protein sequence similarity with OsGA20ox2 between 48.48% to 48.98%. The expression levels of GA20ox9/GA20ox11 in the main stem were significantly higher than those in other tissues. Therefore, it was speculated that GA20ox9/GA20ox11 may be the key genes involved in regulating the development of main stem.

Our results suggested that members of the GA20ox and GA3ox gene families had a certain degree of functional redundancy, but the expression of each gene was spatio-temporal and tissue specific, suggesting their functional diversity. For example, there are five GA20ox genes in Arabidopsis thaliana, in which AtGA20ox1 and AtGA20ox2 are expressed in vegetative growth phase, the former is mainly regulated by biological clock and the latter is mainly regulated by far-red light. AtGA20ox3 is expressed in the outer epidermis, seeds and fruits (Rieu et al., 2008; Phillips et al., 1995), AtGA20ox4 is expressed in roots, and AtGA20ox5 is expressed in fruits (Xu et al., 1995). AtGA3ox1 and AtGA3ox2 are mainly active during germination and vegetative growth, while AtGA3ox3 and AtGA3ox4 are mainly active in reproductive growth (Mitchum et al., 2006). Our results also unveil dynamic expression patterns among AhGA20ox and AhGA3ox genes across various tissues and stages of peanut pod development. Specifically, genes such as AhGA20ox1/AhGA20ox4, AhGA20ox12/AhGA20ox15, AhGA3ox1, and AhGA3ox4/AhGA3ox5 appeared to play potential roles in orchestrating the elongation and growth of peanut pods. Meanwhile, other members within the AhGA20ox and AhGA3ox gene families may contribute to the expansion and material accumulation during the middle and late stages of pod development. This interplay of intricate expression suggested that the functions of AhGA20ox and AhGA3ox genes are not only complementary but also highly spatio-temporally specific. It underscores the notion that the regulation of gibberellin on peanut plant growth and development is a multifaceted process, orchestrated by a network of genes rather than any single entity. To dive deeper into the exact functional roles and regulatory mechanisms of these genes in the context of peanut growth and development, further experimentation is warranted. These experiments will provide a more comprehensive understanding of the nuanced roles and interactions of GA biosynthesis genes in shaping the growth and development of peanut plants.

Conclusions

In this study, we identified 15 GA20ox and five GA3ox family members in peanut, which were distributed on 14 chromosomes, with the exception of chromosomes 1, 6, 10, 11, 16 and 20, and they demonstrated relatively conserved exon-intron patterns mostly with two introns of AhGA20ox genes and one intron of AhGA3ox genes. Phylogenetic analysis divided the AhGA20ox and AhGA3ox into two groups. The conserved pattern of gene structure, promoter cis-elements, and protein motifs further confirmed their evolutionary relationship within the peanut genome. The expression analysis showed that AhGA20oxs and AhGA3oxs were differentially expressed in different tissues and pod of different developmental stages. These findings offer valuable insights into the potential roles of AhGA20ox and AhGA3ox genes in peanut growth and pod development, however, unraveling their precise functional contributions requires further investigation through gene knockout or overexpression studies in the future. The knowledge gained from this study will ultimately facilitate crop improvement strategies and breeding programs aimed at selecting peanut varieties with optimized gibberellin biosynthesis pathways to enhance yield, pod development, and stress tolerance.

Supplemental Information

Table S1 Gene specific primers used in this study

Click here for additional data file.

Table S2 The information of AhGA20ox and AhGA3ox gene family members of peanut, Arabidopsis, rice and soybean

Click here for additional data file.

Table S3 The promoter region of AhGA20ox and AhGA3ox genes

Click here for additional data file.

Table S4 Key cis-acting elements of AhGA20ox and AhGA3ox genes

Click here for additional data file.

Table S5 Chromosomal location data

Click here for additional data file.

Table S6 The Nonsynonymous (Ka)/synonymous (Ks) ratios of the peanut GA20ox and GA3ox genes

Click here for additional data file.

Table S7 Protein-protein interaction results of AhGA20ox and Ah GA3ox members

Click here for additional data file.

Table S8 Heatmap data of GA20ox and GA3ox genes in different pod development stage

Click here for additional data file.

Table S9 Heatmap data of GA20ox and GA3ox genes in different tissues

Click here for additional data file.

Additional Information and Declarations

Competing Interests

Author Contributions

DNA Deposition

Data Availability

The authors declare there are no competing interests.

Jie Sun performed the experiments, analyzed the data, prepared figures and/or tables, authored or reviewed drafts of the article, and approved the final draft.

Xiaoqian Zhang performed the experiments, analyzed the data, prepared figures and/or tables, and approved the final draft.

Chun Fu analyzed the data, prepared figures and/or tables, authored or reviewed drafts of the article, and approved the final draft.

Naveed Ahmad analyzed the data, prepared figures and/or tables, authored or reviewed drafts of the article, and approved the final draft.

Chuanzhi Zhao analyzed the data, authored or reviewed drafts of the article, and approved the final draft.

Lei Hou performed the experiments, analyzed the data, prepared figures and/or tables, and approved the final draft.

Muhammad Naeem analyzed the data, prepared figures and/or tables, authored or reviewed drafts of the article, and approved the final draft.

Jiaowen Pan performed the experiments, analyzed the data, authored or reviewed drafts of the article, and approved the final draft.

Xingjun Wang conceived and designed the experiments, authored or reviewed drafts of the article, and approved the final draft.

Shuzhen Zhao conceived and designed the experiments, prepared figures and/or tables, authored or reviewed drafts of the article, and approved the final draft.

The following information was supplied regarding the deposition of DNA sequences:

The sequences of phylogenetic analysis are available at GenBank (Table S2).

The following information was supplied regarding data availability:

The raw data are available in the Supplemental Files.

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
