# Peer review of "Genome-wide identification and expression analysis of GA20ox and GA3ox genes during pod development in peanut"

_PeerJ, doi:10.7717/peerj.16279_

## Round 0.1 · original submission · Minor Revisions

Dear Authors,
Your manuscript contains valuable information about the functions of important genes involved in GA synthesis in peanuts. The reviewers touched upon the areas where your paper, which has high scientific value, needs improvement. Additionally, you can find my recommendations below.
1. The flow should be arranged by changing the places of the paragraphs in the introduction. Lines 85-96 should be moved to the beginning of the introduction.
2. The first 2 paragraphs that form the beginning of the introduction should be moved after line 84.
3. Lines 99-101 should stay where they are, that is, in the part just before the aim of the study.
4. The following sentences should be deleted.
"In addition, ethylene and ABA were involved in the accumulation of peanut pod at the later stage of development. Taken together, these studies suggested that various hormones regulate the development of peanut pod (Kumar et al., 2019)."
5. The following sentences should be moved to the appropriate places in the Material-Methods and Results section.
"In order to explore the regulation of GA synthesis related genes on peanut pod development, qRT-PCR was performed to examine the expression pattern of AhGA20ox and AhGA3ox genes at different stages of peanut pod development. The results will further enrich the concept of hormonal regulation In addition, the expression patterns of AhGA20ox and AhGA3ox genes in different tissues of peanut cultivar JH8 were also studied by qRT-PCR, which laid a foundation to study the functions of AhGA20ox and AhGA3ox genes."
6. Growing and climatic conditions of the plant should also be specified in the Materials and Methods section, and a relationship should be established between them and GA synthesis should be added to the discussion section.
7. In addition, since the sampling is done at different developmental stages of the plant, information about the GA change in these stages should be added.

Best regards

·

Basic reporting

Line 73-74. citation is required.

The passages between L.85 and L.101 should be the first passage of inroduction. Them aim is the peanut plant.

The aim of this study should be enriched.

The heighighted passage is better to put in methot section.

Experimental design

Line 114. the highlighted cultivar. is it possible to do such a great analysis with a single plant, no population and so on...

Line 116-117. There is no information about RNA extraction and no citation.
Line 119-123. is there any relation between Rice, soybean, and peanut plants gene family?
Line 138. is it possible to mean MEGA 7.0 version software, please pay attention to the highlighted words and if you mean MEGA 7 please give citation to this software?

Line 161-162 please give citation?

Validity of the findings

It is well constructed and well written. The results and discussion section is also very well written. It has always been my priority to bring new and uncultivated plants into the service of humanity and to carry out scientific studies on them. However, this work is admirable.

Additional comments

It is well constructed and well written. The results and discussion section is also very well written. It has always been my priority to bring new and uncultivated plants into the service of humanity and to carry out scientific studies on them. However, this work is admirable.

Reviewer 2 ·

Basic reporting

I suggest editing the English language of the article.
No other comment.

Experimental design

no comment

Validity of the findings

In this study, for the first time, genome-wide characterization and identification of the GA20 oxidase (GA20ox) and GA3 oxidase (GA3ox) in the peanut was performed using bioinformatics tools. Since the functional roles of these gene family members, especially their effects on growth and development and response to stress, need to be understood, expression levels of twenty GA20ox and GA3ox genes containing promotor cis-elements related to pod development and different tissue were determined. These genes exhibited different expression patterns in tissues. As a result of gene expression analysis, the determination of GA20ox and GA3ox genes whose expression levels increase/decrease under different pod development stages will form the basis for a better understanding of the functional mechanisms of GA20ox and GA3ox during development.
However, improvements should be made to the issues I mentioned below in the manuscript.

Additional comments

I find it very necessary to make improvements to the issues I mentioned below;
1- The scientific name of the peanut has not been written anywhere in the manuscript except for keywords. It should be rewritten in the title or introduction section.
2- Why ‘Ah’ codes for the GA20ox and GA3ox genes have been used? How did the authors order these genes (gene numbers)? According to chromosomal location or anything else? This information should be given in the method section.
3- How did the author confirm the GA20ox and GA3ox genes? Which domains have been checked for identifying these proteins?
The authors mentioned that ‘gibberellin-dioxygenases domains were confirmed’ in the results. But they did not give domain names. Absolutely, Pfam domain codes must be mentioned in line 133-135.
4- In the Abstract, results; the genes were randomly distributed across the twenty chromosomes. However, figure 5 shows us the scattered genes on fourteen chromosomes.
5- Line 144, change MRGA to MEGA.
6- On line 161, which website was used for promoter sequences from the peanut genome data? If NCBI used, mention it.
7- miRNA interaction analysis should be part of the manuscript. The author may refer to the suggested papers for the analysis.
8- Evolutionary analysis using Ka/Ks should also be included
9- In line 118, Plant materials and treatment conditions titles must be after bioinformatics analyses, before expression analysis, since this title is related to qPCR analysis.
10- In line 180, a reference should be cited for calculating 2 -ΔΔCT.
11- In Figure 1, the colored ranges for Group B and Group C should be changed; group B with pink, and Group C with blue.
12- In lines 243-246, Because the localizations of all genes are seen on the chromosomal map (Fig 5), there is no need to rewrite them all.
13- In lines 253-255, The sentence should be revised.
14- In lines 251-260. Figure 6 needs to be explained. This figure shows us only protein codes. This section should be written more descriptively by giving protein numbers and protein functions. We should be able to see all the explanations in the figure.
15- In general, it may be more accurate to write the scientific names for the species (rice (Os), soybean (Gm), and peanut (Ah)) In the phylogenetic tree, this situation will be more understandable since the gene codes are given by coding for the first letters of the species names.

For the references;
Salazar et al., 2018 is not in the reference list
Fuentes et al., 2018 but … 2012 in the reference list
Honi et al., 2020; this reference is not relevant in Line 131

Reviewer 3 ·

Basic reporting

In the present manuscript entitled ‘Genome-wide identification and expression analysis of gibberellin synthesis related genes during pod development in peanut’, authors identified a total of fifteen AhGA20ox and five AhGA3ox genes in peanut and characterized them at the molecular level. The authors also assessed the expression of members of AhGA20ox and 5 AhGA3ox families at various pod developmental stages and in different tissues of peanut. The strong expression of AhGA20ox1/ AhGA20ox4, AhGA20ox12/AhGA20ox15, AhGA3ox1, and AhGA3ox4/AhGA3ox5 in the S1-stage indicated that these genes could have a key role in controlling peg elongation and growth. Furthermore, the expression of AhGA20ox and AhGA3ox also suggested a diverse pattern in different peanut tissues including leaves, main stems, flowers, and inflorescences.
Overall, the study indicated the important role of AhGA20ox and AhGA3ox genes in GA-induced mechanisms in peanut plant growth and development which is very informative for researchers, particularly for the peanut scientific community.

Major Comments:

The language quality of the manuscript is very poor and needs to be improved. For instance,
Line 26-27. A whole-genome analysis of the AhGA20ox and AhGA3ox gene families in peanut were identified
Line 234-235: AhGA20ox13 and AhGA3ox4 was positioned at chr.9, while AhGA20ox3 and AhGA20ox4 were found at chr.12.

Line 29: Please replace multiple bioinformatics methods with multiple bioinformatics tools.

Line 32: Phylogenetic analysis divided these members of AhGA20ox and AhGA3ox families into three main groups.
Phylogenetic analysis divided the GA20ox and GA3ox in three groups, but AhGA20ox and AhGA3ox in two groups.

Line 80-82: Further studies have shown that the application of GA, independent of
pollination and fertilization, can promote fruit-setting and paraphyletic outcomes in some crops, as has been demonstrated in pears, apricots, strawberries and grapes.
Similarly, after lines 94-96: GA promoted not only the elongation and growth of peg in the early stage of pod development, but also nutrient accumulation in the middle and late stage of peanut pod development.
Similarly lines 96-97: In addition, ethylene and ABA were involved in the accumulation of peanut pod at the later stage of development.

Please add references.
Some sentences seem to be incomplete such as:
Line no. 138-39: Finally, iTOL (https://itol.embl.de/) online software.

Line 170-171: cyclic reaction (95#, 10s; 60 #, 30 s; 40 cycles) and
Just would like to confirm PCR conditions, is it two step (95C and 60C) or three steps (95C, ?C and 60C).

Please recheck the information related to figure 1 in the manuscript:
Line 192-194: The largest group was group C with 32 members of GA20ox and GA3ox gene family, including all AhGA20ox family members.
As per figure 1, the largest group is B that contains 32 members of only GA20ox. I did not observe in group C both 20ox and 3ox. Please check it.

Line 196: Moreover, most of the genes related to GA synthesis in the same species were clustered together.
As per Fig. 1 GA synthesis genes grouped in mainly two clusters except rice. I observed that 20ox formed one group while 3ox formed another separate group. Please check and describe properly the figure 1.

Line 211-214: Further analysis showed that members of AhGA3ox gene family presented in group B (Fig. 1). In addition, motif 8 exists only in the AhGA3ox gene family (Fig. 3). All members of the AhGA20ox gene family exist in the group C (Fig. 1), while motif 6 and 7 exist only in the AhGA20ox gene family (Fig. 3).

Group B in figure 1 represents AhGA20ox whereas Group C in figure 1 represents AhGA3ox. Please check figure 1
4 and 6 or 6 and 7 ??? please check. It would be better the contrasting colour will be used.
Please check figure 1
Therefore, it can be inferred that motif 6 and 7 are conserved domains specific to the group C
Please check with figure 1

Line 201-204: The exon-intron structure analysis of 15 AhGA20ox and 5 AhGA3ox
202 genes showed that except that AhGA20ox4 had four exons and three introns, the other AhGA20ox genes contain three exons and two introns. Furthermore, AhGA20ox4 had one longest intron. All AhGA3ox genes contain two exons and one intron (Fig. 2).

Please write fifteen AhGA20ox and five AHGA3ox. Moreover, describe properly.

Line 252-253: Peanut pod development was S1divided into six stages including green or purple aerial- grown pegs
Please remove S1.

Line 268-270: AhGA20ox1/AhGA20ox4, AhGA20ox12/AhGA20ox15, AhGA3ox1 and AhGA3ox4/ AhGA3ox5 showed high expression levels in S1 (Fig. 6),
Figure 6 or Figure 8 ??

Line 347-348: Fig. 1 Phylogenetic analysis of GA2ox and GA3ox protein in Arabidopsis Thaliana (At), rice (Os), soybean (Gm) and peanut (Ah).
Please replace: GA2ox with: GA20ox.

Line 349-350: The numbers represent The scale of

Please replace: The scale of with: the scale of.

In Figure 3: Please depict the motif 3 and 6 with different colour, both seems to be same
In Figure 4: I think promoter region is little bit confusing. it should be 0 to -2000 because author is discussing about upstream region. So it can be represented like -2000, -1600, -1200, -800, -400 and 0

In Figure 5: Chromosome mapping of AhGA20ox and AhGA3ox genes.
Fifteen AhGA20ox and 5 AhGA3ox genes were unevenly distributed on the 14 chromosomes,
with the exception of chr. 01, 06, 10, 11 and 16.
What about the 20th chromosome

In Tabl1 1
MV or MW

Experimental design

The authors did not mention internal control for expression assays.

Validity of the findings

no

Additional comments

Manuscript needs further improvements in language

---

## Round 0.2 · Minor Revisions

Please resubmit your manuscript, making the corrections listed by the reviewer, who had requested a major revision in her/his previous review.

Reviewer 2 ·

Basic reporting

no comment

Experimental design

no comment

Validity of the findings

no comment

Additional comments

Here are some of the changes I consider necessary in the manuscript;
Line 101; ‘The Arabidopsis GA20ox and GA3ox gene sequences’ delete ‘gene’, write ‘protein’
Line 101-104; ‘gene sequences of AhGA20oxs and 103 AhGA3oxs were searched against NCBI database using the amino acid sequences as queries, respectively’ This sentence should be changed
‘the amino acid sequences of GA20oxs and GA3oxs of Arabidopsis were searched against peanut genome using NCBI database’.
Line 104; delete ‘respectively’
Line 106; not genes. should be 'domain'. and domain name (2OG-Fe (II) oxygenase domain) and pfam codes should be written here. Because all GA genes were validated according to domains.
Line 106-107; change the sentence as ‘The gene family members (AhGA20ox1 to AhGA20ox15, and AhGA3ox1 to AhGA3ox5) were named and numbered according to peanut species name and the order of chromosome locations from A and B genome, respectively.’
Line 213-216; 20 genes were mapped on the 14 chromosomes. But, the locations of only 12 among them is mentioned why?
Line 224-225; The supplementary table for protein-protein interaction results needs to be given.
Line 234; delete ‘respectively’. Add (Fig. 7).
Line 316; peanut GA20ox and Ga3ox genes were divided into two groups, not three.

Annotated reviews are not available for download in order to protect the identity of reviewers who chose to remain anonymous.

---

## Round 0.3 · Minor Revisions

You are repetitive in the conclusion section. It can be used in the conclusion section from Line 312-321. If you correct it, your manuscript will be better.

---

## Round 0.4 · Minor Revisions

I don't see any semantic changes. The conclusion section should contain recommendations for future work. This sentence "These results provide valuable information for better understanding the roles of GA20ox and AhGA3ox genes in peanut growth and pod development" seems unfinished to me.
"Therefore, more experiments are needed to further study the exact function of these genes and the gene regulation mechanism in peanut." I'm again asking to understand why you insist on keeping your sentence in the discussion section.

---

## Round 0.5 · accepted · Accept

Thank you for your revision. I am pleased to inform you that your manuscript has been accepted.